# Oral Administration of *Lactobacillus rhamnosus* Ameliorates the Progression of Osteoarthritis by Inhibiting Joint Pain and Inflammation

**DOI:** 10.3390/cells10051057

**Published:** 2021-04-29

**Authors:** JooYeon Jhun, Keun-Hyung Cho, Dong-Hwan Lee, Ji Ye Kwon, Jin Seok Woo, Jiyoung Kim, Hyun Sik Na, Sung-Hwan Park, Seok Jung Kim, Mi-La Cho

**Affiliations:** 1Rheumatism Research Center, Catholic Research Institute of Medical Science, Catholic University of Korea, Seoul 06591, Korea; jhunjy@catholic.ac.kr (J.J.); chokh@catholic.ac.kr (K.-H.C.); jyyj7082@nate.com (J.Y.K.); ulbojs@catholic.ac.kr (J.S.W.); jykim0727@catholic.ac.kr (J.K.); nayoy@catholic.ac.kr (H.S.N.); 2Department of Biomedicine & Health Sciences, College of Medicine, Catholic University of Korea, 222, Banpo-daero, Seocho-gu, Seoul 06591, Korea; 3Department of Orthopedic Surgery, Uijeongbu St. Mary’s Hospital, College of Medicine, Catholic University of Korea, Seoul 06591, Korea; ldh850606@naver.com; 4Division of Rheumatology, Department of Internal Medicine, Seoul St. Mary’s Hospital, College of Medicine, Catholic University of Korea, Seoul 06591, Korea; rapark@catholic.ac.kr; 5Department of Medical Lifescience, College of Medicine, Catholic University of Korea, 222, Banpo-daero, Seocho-gu, Seoul 06591, Korea

**Keywords:** osteoarthritis, monosodium iodoacetate (MIA), inflammation, microbiota, *Lactobacillus rhamnosus*

## Abstract

Osteoarthritis (OA) is the most common form of arthritis and age-related degenerative joint disorder, which adversely affects quality of life and causes disability. However, the pathogenesis of OA remains unclear. This study was performed to examine the effects of *Lactobacillus rhamnosus* in OA progression. OA was induced in 6-week-old male Wistar rats by monosodium iodoacetate (MIA) injection, and the effects of oral administration of *L. rhamnosus* were examined in this OA rat model. Pain severity, cartilage destruction, and inflammation were measured in MIA-induced OA rats. The small intestines were isolated from OA rats, and the intestinal structure and inflammation were measured. Protein expression in the dorsal root ganglion was analyzed by immunohistochemistry. The effects of *L. rhamnosus* on mRNA and protein expression in chondrocytes stimulated with interleukin (IL)-1β and lipopolysaccharide (LPS) were analyzed by real-time polymerase chain reaction (RT-PCR) and enzyme-linked immunosorbent assay (ELISA). Pain severity was decreased in *L. rhamnosus*-treated MIA-induced OA rats. The levels of expression of MCP-1, a potential inflammatory cytokine, and its receptor, CCR2, were decreased, and GABA and PPAR-γ expression were increased in *L. rhamnosus*-treated OA rats. The inflammation, as determined by IL-1β, and cartilage destruction, as determined by MMP3, were also significantly decreased by *L. rhamnosus* in OA rats. Additionally, intestinal damage and inflammation were improved by *L. rhamnosus.* In human OA chondrocytes, TIMP1, TIMP3, SOX9, and COL2A1 which are tissue inhibitors of MMP, and IL-10, an anti-inflammatory cytokine, were increased by *L. rhamnosus*. *L. rhamnosus* treatment led to decreased pain severity and cartilage destruction in a rat model of OA. Intestinal damage and inflammation were also decreased by *L. rhamnosus* treatment. Our findings suggested the therapeutic potential of *L. rhamnosus* in OA.

## 1. Introduction

Osteoarthritis (OA) is the most common form of arthritis in older adults and is associated with persistent pain that gradually progresses with repeated alternating periods of improvement and deterioration. Therefore, OA is considered an important treatment target in modern society, with an emphasis on quality of life (QOL). OA has long been understood as a disease that progresses due to the mechanical “wear and tear” of cartilage, but this view has changed due to the development of molecular biology in the 1990s [1]. Recent studies have shown that OA has an inflammatory pathogenesis involving numerous proinflammatory cytokines and matrix metalloproteinases (MMPs), leading to cartilage destruction [2,3]. However, unlike rheumatoid arthritis (RA), which responds well to anticytokine therapy, such treatment is not effective for OA, suggesting that more diverse methods of OA treatment with multiple targets are needed [3]. There has been increasing research interest in the microbiome, and studies have shown that the worsening of OA can be prevented by modulation of the gut microbiota based on the concepts of the gut–joint axis, cartilage–gut–microbiome axis, etc. [4,5]. A method of controlling chronic systemic low-grade inflammation through changes in gut microbiota has been proposed as a new therapeutic option for OA [5]. Probiotics are being actively studied to provide health benefits in the host by changes in the gut microbiota. Probiotics have been shown to improve symptoms of diarrhea, bacterial infection, atopic dermatitis, inflammatory bowel disease (IBD), and autoimmune disorders, such as RA, as well as in an experimental model of multiple sclerosis [6,7]. The immunomodulatory effect is known to vary among various probiotics strains depending on the administration method, the type of immune disorder, and the severity of the disease [7]. Several studies have examined the effects of specific probiotic strains applied to OA. Although there have been fewer such studies related to OA in comparison to RA or IBD, the effect has been confirmed in several strains. *Lactobacillus casei* strain Shirota, *Lactobacillus acidophilus*, and *Streptococcus thermophilus* were applied to experimental models of OA and clinically applied in patients and shown to be effective [8,9,10,11].

*Lactobacillus rhamnosus* is known to produce butyrate, a short-chain fatty acid (SCFA), through alteration of the gut microbiota. Therefore, *L. rhamnosus* may have an immunomodulatory effect and a protective effect on cartilage [12,13], suggesting that *L. rhamnosus* may be another probiotic strain with a therapeutic effect on OA. To confirm its therapeutic effects, *L. rhamnosus* (LR-2) was administered to a rat monosodium iodoacetate (MIA)-induced OA model, and its antinociceptive and anti-inflammatory effects were examined by confirming the immunomodulatory mediators and catabolic/anabolic factors through histological and immunohistochemical (IHC) analyses.

To confirm that the administration of probiotics has an immunomodulatory effect and a protective effect on cartilage through changes in the intestinal environment, the small intestine was examined by staining with hematoxylin and eosin, and IHC staining after administering *L. rhamnosus* (LR-2) to the rat OA model. In addition, *L. rhamnosus* (LR-2) was shown to regulate the levels of inflammatory mediators and anabolic factors in human chondrocytes.

We used the heat-killed *L. rhamnosus* throughout the process of this experiment. A number of studies proved that heat-killed bacteria have several benefits compared to live bacteria, such as a longer validity and guaranteed safety due to the absence of infection risk. There is also no loss of activity when used in conjunction with antibiotics or antifungal agents [14,15]. It has been proved that the heat-killed bacteria not only has these benefits, but also shows similar effects to live bacteria [16,17]. There were predictions that the viability of the anaerobic bacteria, *Lactobacillus rhamnosus* would not be constant in the process of delivery to mice after being prepared in a live bacteria form. In this situation, similar functions and mechanisms of live bacteria were reported for heat-killed bacteria. And there is report that the heat-killed *L. rhamnosus* and live *L. rhamnosus* show similar effects. So, we injected heat-killed *Lactobacillus rhamnosus* into MIA rats [18,19]. Also, the heat-killed *L. rhamnosus* is easier to use in clinical studies for individuals because of its safety guarantee. By using heat-killed *L. rhamnosus* in this study, we may be able to present a stronger theoretical basis in future clinical studies.

## 2. Materials and Methods

### 2.1. Ethics Statement

The Animal Care Committee of The Catholic University of Korea approved the experimental protocol (permit number: CUMC-2020-0037-02). All animal-handling procedures and protocols followed the guidelines of the Animal Research Ethics Committee of the Catholic University of Korea and the US National Institutes of Health Guidelines. All relevant protocols were approved by the institutional review board of Uijeongbu St. Mary’s Hospital (HC14TISI0071) and performed in accordance with the Declaration of Helsinki. All patients provided written, informed consent.

### 2.2. Induction of OA in Rats and Treatment with L. rhamnosus (LR-2)

The animals were randomly assigned to treatment groups prior to the beginning of the study. Rats were divided into two groups (6 rats in each group) and were group-housed in two cages. Each in vivo experiment was repeated a total 3 times. A total of 36 rats were used in this study. After anesthetization with isoflurane, rats were injected with 3 mg of MIA (Sigma-Aldrich, St. Louis, MO, USA) in a volume of 50 µL using a 26.5-G needle inserted through the patellar ligament into the intraarticular space of the right knee; *L. rhamnosus* (LR-2) was suspended in phosphate-buffered saline (PBS) at 500 mg/kg and killed by heating at 80 °C for 30 min. *L. rhamnosus* (LR-2) and vehicle (saline) were administrated orally every day for 28 days after MIA induction. The rats were sacrificed on day 28 after MIA injection.

### 2.3. Assessment of Pain Behavior

MIA-treated rats were randomized into the experimental groups. Osteoarthritis was induced in 6-week-old male Wistar rats (*n* = 6). The experiment was performed three times for each paw. Nociceptive testing was performed using a dynamic plantar aesthesiometer (Ugo Basile, Gemonio, Italy), which uses an automated version of the von Frey hair assessment procedure, before MIA injection on day 0, and once a week thereafter [20]. The rats were placed on a metal mesh surface in an acrylic chamber in a temperature-controlled room (21 °C–22 °C) and allowed to acclimatize for 15 min before testing. The touch stimulator unit was placed beneath the animal, and an adjustable angled mirror was used to insert the stimulating microfilament (0.5 mm in diameter) below the plantar surface of the hind paw. When the instrument was activated, a fine plastic monofilament was advanced at a constant speed and touched the paw in the proximal metatarsal region. The filament exerted a gradually increasing force on the plantar surface, starting below the threshold of detection, and increasing until the stimulus became painful, which was indicated by withdrawal of the paw. The force required to elicit a paw withdrawal reflex was recorded automatically and measured. A maximum force of 50 g and ramp speed of 20 s were used for all esthesiometry tests. Behavioral tests of secondary tactile allodynia were conducted immediately before administration of *L. rhamnosus* (LR-2).

### 2.4. Weight-Bearing Measurement

Weight bearing was evaluated using an incapacitance tester (Linton Instrumentation, Norfolk, UK) that included a dual-channel weight mean value. The rats were attentively positioned in a plastic chamber. The strength applied by an individual hind limb was averaged over more than a 3-s time. The individual data point was the average of three measurements. The percentage of weight divided onto the handled (ipsilateral) hind limb was calculated utilizing the following equation: (weight on right leg/weight on right leg and left leg) × 100.

#### 2.4.1. Histological and Immunohistochemical Analyses

Histological changes were assessed to determine the effects of *L. rhamnosus* (LR-2) treatment in the knee joint, small intestine, and dorsal root ganglion of rats. The animals were perfused via the ascending aorta with 10% neutral buffered formalin (pH 7.4). The knee joints, including the patella and joint capsule, were resected and maintained in the same fixative for an additional 48 h at 4 °C. The fixed knee joint specimens were decalcified with 5% formic acid for 6 days at 4 °C, and then embedded in paraffin. Standardized 7-mm serial sections were obtained at the medial and lateral midcondylar level in the sagittal plane and stained with hematoxylin and eosin (H&E), safranin O-fast green, and toluidine blue to enable evaluation of proteoglycan content. A modified Mankin score was used to classify histological injury of the articular cartilage, as follows: 0 = normal; 1 = irregular surface, including fissures into the radial layer; 2 = pannus; 3 = absence of superficial cartilage layers; 4 = slight disorganization (cellular row absent, some small superficial clusters); 5 = fissure into the calcified cartilage layer; and 6 = disorganization (chaotic structure, clusters, and osteoclast activity). Cellular abnormalities were scored on a scale ranging from 0 to 3, where 0 = normal; 1 = hypercellularity, including small superficial clusters; 2 = clusters; and 3 = hypocellularity. Matrix staining was scored on a scale ranging from 0 to 4, where 0 = normal/slight reduction in staining; 1 = staining reduced in the radial layer; 2 = staining reduced in the interterritorial matrix; 3 = staining present only in the pericellular matrix; and 4 = staining absent. Joint space width was estimated based on the sum of the nearest distance of the medial and lateral tibiofemoral joints. Formalin-fixed small intestine and dorsal root ganglion were embedded in paraffin and stained by IHC or hematoxylin and eosin (H & E). Histological evaluations were performed independently by two experienced researchers who were blinded to the study groups.

#### 2.4.2. Immunohistochemistry

Sections were deparaffinized and rehydrated using a graded ethanol series and incubated overnight at 4 °C with antibodies to interleukin (IL)-1β (Santa Cruz Biotechnology, Santa Cruz, CA, USA), matrix metalloproteinase-3 (MMP-3; Abcam, Cambridge, UK), tissue inhibitor of metalloproteinase-3 (TIMP-3) (Abcam), Interleukin-10 (IL-10) (Abcam), CCR2 (Novus Biologicals, Littleton, CO, USA), GABA (Novus Biologicals), and monocyte chemoattractant protein (MCP)-1 (Abcam). The slides were then treated with secondary antibodies and biotinylated anti-mouse IgG for 20 min conjugated to streptavidin peroxidase complex (Vector Laboratories, Burlingame, CA, USA) for 1 h, and then treated with 3,30-diaminobenzidine (Dako, Glostrup, Denmark). The slides were counterstained with Mayer’s hematoxylin and photographed under a photomicroscope (Olympus, Tokyo, Japan). Immunohistochemistry evaluations were performed independently by two experienced researchers who were blinded to the study groups. The positive cell percentage was analyzed with HDAB (hematoxylin and DAB) by selecting color deconvolution in the plugin item in the image J program (NIH, MD, USA).

#### 2.4.3. Primary Culture and Treatment of OA Chondrocytes

All relevant protocols were approved by the institutional review board of Uijeongbu St. Mary’s Hospital (HC14TISI0071) and performed in accordance with the Declaration of Helsinki. All patients provided written informed consent. OA was diagnosed using the American College of Rheumatology criteria [21]. To isolate chondrocytes, we obtained knee-joint cartilage samples from OA patients during joint-replacement surgery. Chondrocytes were obtained by digesting the articular cartilage. The experiments were performed with cartilage from 3 individuals in each experiment (sex: male 2, female 1, age 73 ± 3). Cartilage samples from OA patients were washed in calcium- and magnesium-free PBS and finely ground. Chondrocytes were obtained by digesting the articular cartilage with 0.2% pronase (Sigma) for 1 h, followed by digestion with 0.2% clostridia collagenase (Sigma) for 3 h at 37 °C in high-glucose Dulbecco’s modified Eagle’s medium (DMEM; Life Technologies, Carlsbad, CA, USA) containing an antibiotic–antimycotic solution (100 U/mL penicillin, 100 μg/mL streptomycin, and 0.25 μg/mL amphotericin B; Life Technologies). Undigested cartilage was removed with a 70-μm nylon mesh (cell strainer; Falcon Plastics, Oxnard, CA, USA), and the chondrocytes were collected by centrifugation. Cells were then washed twice, followed by resuspension in DMEM supplemented with 10% fetal bovine serum (FBS; Life Technologies). Finally, the cells were plated in 100-mm tissue culture dishes for expansion at 37 °C in a humidified 5% CO_2_ atmosphere for 10 days (Shel Lab, Cornelius, OR, USA). Culture medium was changed every 2-3 days. Following expansion, chondrocytes were cultured in FBS-free DMEM (5%, *v*/*v*) and were used as confluent monolayers for all experiments. Cells (1 × 10^5^ cells/well) were plated in 24-well tissue culture plates, and the medium was replaced with serum-free DMEM the following day. Twenty-four hours later, the cells were pretreated with heat-killed *L. rhamnosus* (LR-2) for 2 h and then stimulated with or without recombinant human IL-1β (20 ng/mL: R & D Systems) for 48 h.

#### 2.4.4. Real-Time Polymerase Chain Reaction (RT-PCR)

Total RNA was extracted using TRI Reagent (Molecular Research Center, Cincinnati, OH, USA) according to the manufacturer’s instructions. Complementary DNA (cDNA) was prepared by reverse transcription of single-stranded RNA using a high-capacity cDNA reverse transcription kit (Applied Biosystems, Foster City, CA, USA) according to the manufacturer’s instructions. PCR amplification was performed using a LightCycler 2.0 instrument (software version 4.0; Roche Diagnostics, Indianapolis, IN, USA). All reactions were performed using LightCycler FastStart DNA Master SYBR Green I (TaKaRa, Shiga, Japan) according to the manufacturer’s directions. The primer pairs used were as follows: β-actin, forward: 5′-GGA CTT CGA GCA AGA GAT GG-3′, reverse: 5′-TGT GTT GGG GTA CAG GTC TTT G-3′; TIMP1, forward: 5′-AAT TCC GAC CTC GTC ATC AG-3′, reverse: 5′-TGC AGT TTT CCA GCA ATG AG-3′; TIMP3, forward: 5′-CTG ACA GGT CGC GTC TAT GA-3′, reverse: 5′-GGC GTA GTG TTT GGA CTG GT-3′; SOX9, forward:5′-ACT TGC ACA ACG CCG AG-3′, reverse: 5′-CTG GTA CTT GTA ATC CGG GTG-3′;COL2A1, forward: 5′-TCT ACC CCA ATC CAG CAA AC-3′, reverse: 5′-GTT GGG AGC CAG ATT GTC AT-3′.

#### 2.4.5. Statistical Analysis

Statistical analyses were performed using the nonparametric Mann–Whitney *U* test for comparisons between two groups, and one-way ANOVA with Bonferroni’s post-hoc test for multiple comparisons. GraphPad Prism (version 5.01; GraphPad Software Inc., San Diego, CA, USA) was used for all analyses. The data are presented as the mean ± standard deviation (SD). In all analyses, *p* < 0.05 was taken to indicate statistical significance.

## 3. Results

### 3.1. L. rhamnosus (LR-2) Suppresses Pain in MIA-Induced OA Rats

Paw withdrawal threshold (PWT), paw withdrawal latency (PWL), and weight bearing were examined to investigate whether oral administration of *L. rhamnosus* (LR-2) can control pain in MIA-induced OA rats. The group that was administered *L. rhamnosus* (LR-2) had higher PWT and PWL than the vehicle group (Figure 1A). These results indicated the reduction of pain by administration of *L. rhamnosus* (LR-2). In addition, weight-bearing measurement, which measures the load of both feet confirmed that the left and right loads differ significantly between the vehicle group and the *L. rhamnosus*-administered group (LR-2) and (Figure 1B). This indicated that the pain was reduced by administration of *L. rhamnosus* (LR-2). These results suggest that *L. rhamnosus* treatment improved weight bearing, PWL, and PWL in rats with MIA-induced OA. In IHC analysis of the dorsal root ganglion (DRG), the *L. rhamnosus* (LR-2)-administered group showed higher levels of expression of PPAR-¦Ã and GABA, which are known to control pain, compared to the vehicle group. On the other hand, the expression levels of MCP-1 and its receptor, CCR2, which is expected to intensify nervous pain, were reduced in the *L. rhamnosus* (LR-2)-administered group (Figure 2). These results indicated that the administration of *L. rhamnosus* (LR-2) controls OA-induced pain.

### 3.2. Protective Effects of L. rhamnosus (LR-2) against Cartilage Destruction in MIA-Induced OA Rats

Histological analysis was performed using H&E- and safranin O-staining to confirm the degree of cartilage destruction in MIA-induced OA rats. Both the OARSI score and the total Mankin score were low in the *L. rhamnosus* (LR-2)-administered group (Figure 3A,B). In addition, the structure, cells, safranin O-staining level, and tidemark integrity, which are detailed indicators of the total Mankin score, showed low scores in the *L. rhamnosus* (LR-2)-administered group (Figure 3C). These results indicated that administration of *L. rhamnosus* (LR-2) inhibited OA progression.

### 3.3. L. rhamnosus (LR-2) Modulates the Levels of Inflammatory Mediators and Catabolic/Anabolic Factors in the Synovium of MIA-Induced OA Rats

IHC staining was performed on the joint synovium of MIA-induced OA rats to confirm the expression of inflammatory mediators and catabolic/anabolic factors. In comparison with the vehicle group, the expression level of IL-1β showed a decreasing trend in the *L. rhamnosus* (LR-2)-administered group, whereas the expression of the anti-inflammatory cytokine, IL-10, was increased (Figure 4A). In addition, the expression level of MMP3, which is involved in cartilage destruction, was decreased, while the expression of TIMP3, which inhibits MMP3, was increased in the *L. rhamnosus* (LR-2) -administered group (Figure 4B). These results showed that *L. rhamnosus* (LR-2) has a protective effect against cartilage destruction through its inflammatory/anti-inflammatory effects and regulation of catabolic/anabolic responses.

### 3.4. L. rhamnosus (LR-2) Regulates Intestinal Inflammation in OA

To confirm that *L. rhamnosus* (LR-2) changes the intestinal environment and, therefore, has an immunomodulatory effect, we performed H&E and IHC staining in the small intestine. The results indicated that the destruction of the architecture of small intestinal villi was attenuated in the *L. rhamnosus* (LR-2)-administered group (Figure 5A). In addition, IHC staining showed that the expression levels of MCP-1, CCR2, and IL-6 were decreased in the *L. rhamnosus* (LR-2)-administered group compared to the vehicle group. Whereas the expression of the anti-inflammatory cytokine, IL-10, was increased (Figure 5B). These results showed that *L. rhamnosus* (LR-2) has an immunomodulatory effect involving the intestinal immune system and environment.

### 3.5. L. rhamnosus (LR-2) Regulates the Levels of Inflammatory Mediators, Anabolic Factors and Chondrogenic Transcription Factors in Chondrocytes

Human OA chondrocytes were stimulated with IL-1β or LPS together with *L. rhamnosus* (LR-2); and then the transcript levels of anabolic factors TIMP1 and 3; the anti-inflammatory cytokine IL-10; and chondrogenic transcription factors SOX9, and COL2A1 were determined by real-time PCR. The transcript levels of TIMP1, 3, and IL-10 were increased in the *L. rhamnosus* (LR-2)-administered group (Figure 6A–C), consistent with the results of IHC analysis. Also, the transcripts levels of SOX9 and COL2A1 were upregulated in the *L. rhamnosus* (LR-2)-administered condition (Figure 6D). In addition, human chondrocytes were stimulated with LPS along with *L. rhamnosus* (LR-2), and MCP-1 expression was examined by ELISA (Figure 6E). *L. rhamnosus* (LR-2) decreased the expression of MCP-1 compared to vehicle or celecoxib confirming that *L. rhamnosus* (LR-2) has a protective effect in the cartilage.

## 4. Discussion

*L. rhamnosus* is used as an anti-inflammatory supplement, but its therapeutic efficacy in OA is unclear. Our results suggest that *L. rhamnosus* improves OA severity by downregulating cartilage destruction and inducing the expression of anabolic factor and chondrogenic transcription factors. Also, intestinal damage and inflammation were also decreased by *L. rhamnosus* treatment.

With progress in research on the microbiome and the gradual elucidation of the relations between the gut microbiota and various diseases, there has been increasing interest in both basic and clinical research regarding the treatment of various diseases using probiotics. Several of these studies have demonstrated the effectiveness of probiotics in the treatment of various diseases. Treatment with probiotics has been shown to have therapeutic effects in infectious diseases, autoimmune conditions, allergies, and behavior disorders, including Alzheimer’s disease, through changes in the intestinal environment evoked by the gut microbiota, and it is under discussion as to whether probiotics can be applied as an adjuvant therapy in the treatment of COVID-19 [22]. There have been several studies on therapeutic effects of probiotics in RA, an inflammatory and autoimmune disease, and investigations are still ongoing. In contrast, there have been insufficient studies on the therapeutic effects of probiotics in OA. OA can be seen as a form of chronic low-grade inflammation, and the therapeutic effect of probiotics can be expected through the gut–joint axis concept; therefore, further research regarding this issue is needed.

Several strains of probiotics have been studied to date in osteoarthritis. Studies of *L. casei* strain Shirota revealed its effects and mechanism of action in experimental OA. The clinical efficacy of this organism in actual patients, as compared to a placebo control group, was also investigated. Oral administration of *L. casei* Shirota decreases hs-CRP in OA patients’ serum [9]. Another study demonstrated the efficacy of applying *L. acidophilus* in MIA-induced OA. This study results that *L. acidophilus* can alleviate OA-associated pain and delay the progression of the disease by inhibition proinflammatory cytokine production and deducing cartilage damage [10]. The effects of *S. thermophilus* have been studied in experimental OA, and in actual patients. Oral intake of Streptococcus thermophiles improves knee osteoarthritis degeneration [11].

Several recent reports have described the therapeutic effects of the administration of probiotics and prebiotics. Although the detailed mechanisms of action have not been identified, these studies showed that alteration of the gut microbiota affects the production of short chain fatty acids (SCFAs) and other metabolites, mediating anti-inflammatory effects and changes in gut permeability, resulting in therapeutic effects [23,24,25]. It has been reported that SCFAs, acetate, butyrate, and propionate have anti-inflammatory effects mediated through G protein-coupled receptors (GPCRs) or through reduction of histone deacetylases (HDACs) [26]. In addition, it is well-known that SCFAs are involved in the activity of several types of immune cells, and it has been reported that SCFAs play important roles in the differentiation of Treg cells. Butyrate is known to determine the number and function of Treg cells in the intestine [13], and to prevent cartilage degradation by reducing type II collagen degradation [27]. *L. rhamnosus* is known to have an immunomodulatory effect through changes in the intestinal environment and various effectors, and to affect bone formation and chondrocyte differentiation. *L. rhamnosus* is known to produce metabolites, such as lactate and acetate, in the intestine, and acetate is converted to acetyl-CoA through butyrate-producing bacteria resulting in the production of butyrate [12,28,29]. When the degree of inflammation is severe in OA patients, the composition of *Streptococcus* spp. increases and the composition of *Lactobacillus* spp. decreases [30,31]. *L. rhamnosus* is a strain that is widely used and has already been applied in various inflammatory diseases, including RA, to confirm its therapeutic effect [32,33].

Therefore, we performed animal experiments to examine whether *L. rhamnosus* (LR-2) has a therapeutic effect against OA through an anti-inflammatory action mediated via butyrate production. As stated above in the introduction, we used heat-killed *L. rhamnosus* in our experiment. Previous studies revealed that the heat-killed bacteria promotes butyrate production in vitro and in vivo [34] Therefore, we found no difficulty in using the heat-killed *L. rhamnosus* to prove our hypothesis. Administration of *L. rhamnosus* (LR-2) increased the PWT, PWL, and weight bearing capacity of MIA-induced OA rats, and the levels of PPAR-γ and GABA expression in the dorsal root ganglion (DRG) were elevated. On the other hand, the expression levels of MCP-1 and CCR2 were reduced, suggesting that pain severity was reduced through the nociceptive pathway [35]. Further molecular studies are required to understand whether each interaction needs factors.

The evaluation of cartilage destruction is an important index for the progression of OA. In this study, both the OARSI score and total Mankin score on H&E staining of the knee joint were lower after *L. rhamnosus* (LR-2) administration than in the vehicle controls, indicating the suppression of OA progression. In IHC analysis of the joint synovium, the group administered *L. rhamnosus* (LR-2) showed decreased levels of the inflammatory cytokine IL-1β, and the catabolic factor MMP3; and increased levels of the anti-inflammatory cytokine IL-10, and the anabolic factor TIMP3. These observations suggested that when *L. rhamnosus* (LR-2) was administered, cartilage destruction was inhibited through the regulation of inflammatory/anti-inflammatory effects and catabolic/anabolic responses [36,37]. In addition, after administration of *L. rhamnosus* (LR-2) to IL-1β-stimulated human chondrocytes, the expression levels of the anabolic factors, TIMP1, 3, and IL-10, were confirmed to increase, indicating that cartilage destruction was inhibited through the regulation of catabolic/anabolic responses. Furthermore, structural changes in the small intestine after administration of *L. rhamnosus* (LR-2) were confirmed by H&E staining, and the expression levels of MCP-1 and CCR2, chemokines that induce inflammation and tissue injury, were shown to be decreased by IHC staining in the *L. rhamnosus* (LR-2) group. Also, IL-6 was decreased but IL-10 was significantly increased compared to the vehicle treated group. These results support the cartilage protective effect of *L. rhamnosus* (LR-2) mediated through changes in the intestinal immune system and environment.

In our results, osteoarthritis was induced by MIA, and intestinal tissue was damaged. Also, it was confirmed that the damage was recovered again by the *L. rhamnosus* oral administration group. Currently, there have been several studies on the gut–joint axis. Dysbiosis of the bacteria communities can cause many chronic diseases, such as inflammatory rheumatoid arthritis, bowel disease, obesity, cancer, and autism. However, there is no research yet on cases of intestinal problems and dysbiosis when arthritis occurs. First of all, the MIA-induced OA model is an acute inflammatory model.

It is probably thought that pain and inflammatory factors cause intestinal damage. However, there is no research on this mechanism yet. Further investigations of the mechanism of action are still required.

An important limitation of this study is the lack of evidence in humans. The results of this study showed that *L. rhamnosus* (LR-2) protects cartilage through regulation of inflammatory/anti-inflammatory effects and catabolic/anabolic responses in MIA-induced OA. This suggests the possibility of using *L. rhamnosus* (LR-2) as a therapeutic agent in OA patients. Further detailed investigations of the mechanism of action are still required, along with clinical studies.

Furthermore, it is necessary to discover additional probiotic strains with therapeutic effects and to clarify the mechanism of action of each strain. Therefore, it will be necessary to examine whether different strains act via the same therapeutic mechanism, whether a greater therapeutic effect can be achieved due to synergistic effects when two or more strains are administered in combination, or whether there are no differences from the administration of a single strain. Although further studies are required, based on the results presented here, it is expected that probiotics will be useful as pharmabiotics soon.

## Figures and Tables

**Figure 1 cells-10-01057-f001:**
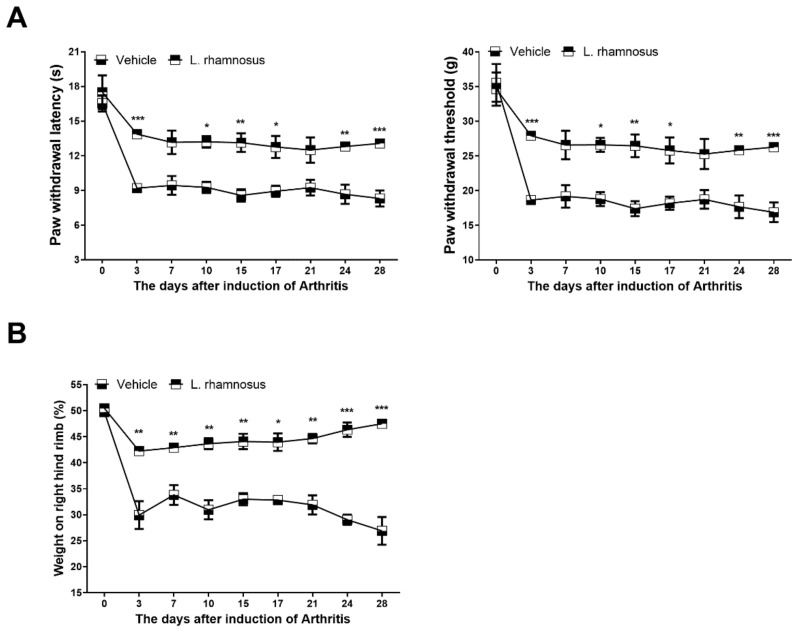
*L. rhamnosus* (LR-2) showed a therapeutic effect in OA rats. (**A**) Pain behavior was analyzed as PWL (left) and PWT (right) in vehicle-treated MIA-induced OA rats, and *L. rhamnosus* (LR-2)-treated MIA-induced OA rats (*n* = 6 per group) until day 28. (**B**) Weight bearing was examined in all groups (*n* = 6 per group) until day 28. Data are presented as the mean ± SD of three independent experiments. * *p* < 0.05, ** *p* < 0.01, and *** *p* < 0.001.

**Figure 2 cells-10-01057-f002:**
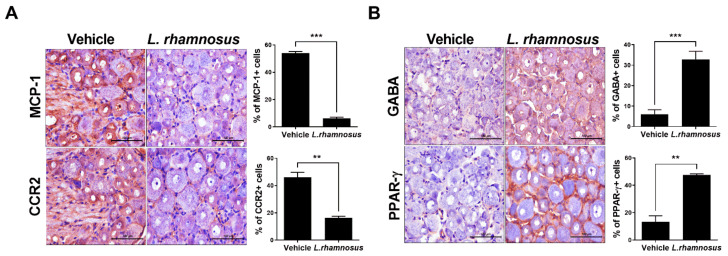
(**A**,**B**) Representative images of immunohistochemical staining for MCP-1, CCR2, GABA, and PPAR-γ in the DRG from vehicle-treated MIA-induced OA rats and *L. rhamnosus* (LR-2)-treated MIA-induced OA rats. Bar graphs show averaged percentages of MCP-1-, CCR2-, GABA-, and PPAR-γ-positive cells in the DRG. Data are presented as the mean ± SD of three independent experiments. ** *p* < 0.01 and *** *p* < 0.001.

**Figure 3 cells-10-01057-f003:**
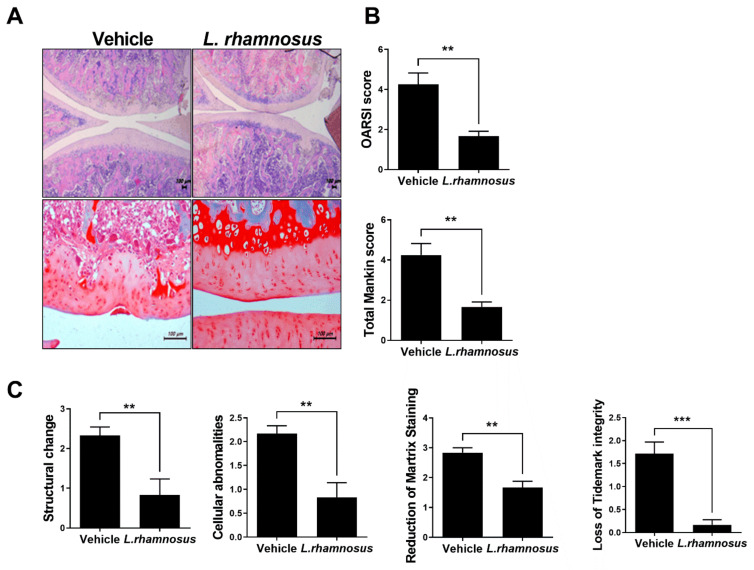
(**A**) Representative images of H&E- and safranin O-stained joints of vehicle-treated MIA-induced OA rats and *L. rhamnosus* (LR-2)-treated MIA-induced OA rats. (**B**) Bar graphs show average OARSI score and Mankin score. (**C**) Bar graphs show structure, cells, staining, and tidemark integrity in vehicle-treated MIA-induced OA rats and *L. rhamnosus* (LR-2)-treated MIA-induced OA rats. Data are presented as the mean ± SD of three independent experiments. ** *p* < 0.01, and *** *p* < 0.001.

**Figure 4 cells-10-01057-f004:**
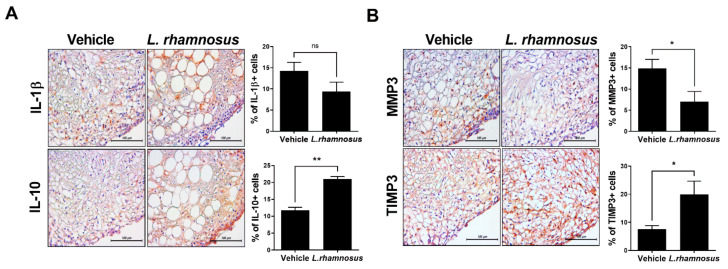
(**A**) Representative images of immunohistochemical staining for IL-1β and IL-10 in joint synovium of vehicle-treated MIA-induced OA rats and *L. rhamnosus* (LR-2)-treated MIA-induced OA rats. Bar graphs show average numbers of IL-1β- and IL-10-positive cells in the joint synovium. (**B)** Representative images of immunohistochemical staining for MMP3 and TIMP3 in joint synovium of vehicle-treated MIA-induced OA rats and *L. rhamnosus* (LR-2)-treated MIA-induced OA rats. Bar graphs show average numbers of MMP3-, and TIMP3-positive cells in the joint synovium. Data are presented as the mean ± SD of three independent experiments. * *p* < 0.05, ** *p* < 0.01. ns = not significant.

**Figure 5 cells-10-01057-f005:**
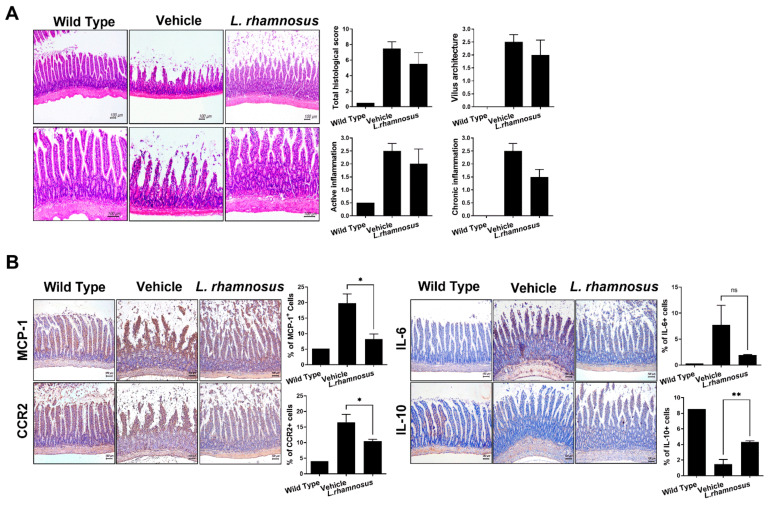
(**A**) Representative images of H&E-stained small intestine from non-OA rats (WT), vehicle-treated MIA-induced OA rats, and *L. rhamnosus* (LR-2)-treated MIA-induced OA rats. Bar graphs show histological score, villus architecture, active inflammation, and chronic inflammation. (**B**) Representative images of immunohistochemical staining for MCP-1, CCR2, IL-6, and IL-10 in the small intestine from non-OA rats (WT), vehicle-treated MIA-induced OA rats, and *L. rhamnosus* (LR-2)-treated MIA-induced OA rats. Bar graphs show average percentages of MCP-1, CCR2, IL-6, and IL-10-positive cells. Data are presented as the mean ± SD of three independent experiments. * *p* < 0.05 and ** *p* < 0.01. ns = not significant.

**Figure 6 cells-10-01057-f006:**
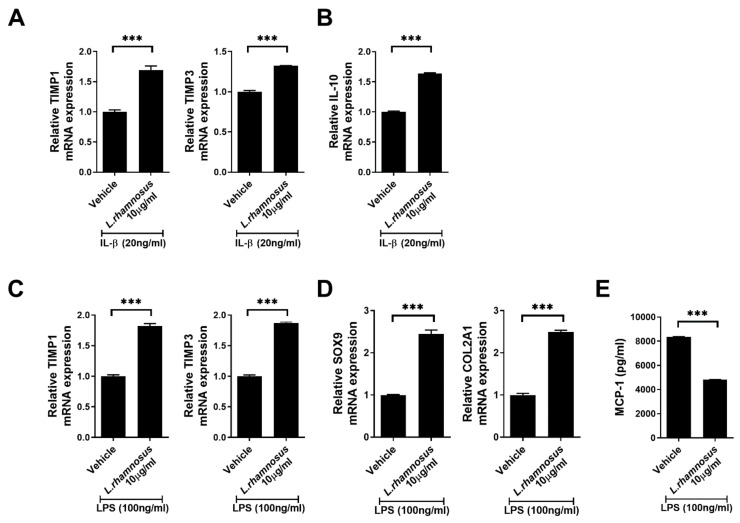
Human articular chondrocytes from OA patients were cultured with IL-1β and LPS in the presence or absence of *L. rhamnosus* (LR-2) for 48 h, following 24 h in serum-free medium. The mRNA expression levels of anabolic factors, tissue inhibitors of (**A**) metalloproteinase (TIMP)-1, TIMP-3, and (**B**) IL-10 were measured by quantitative real-time PCR; β-actin was used as an internal control. The transcripts expression levels of anabolic factors TIMP-1 and TIMP3 (**C**), chondrogenic transcription factors SOX9 and COL2A1 (**D**). (**E**) The concentrations of MCP-1 were measured by enzyme-linked immunosorbent assay. Data are presented as the mean ± SD of three independent experiments. *** *p* < 0.001.

## Data Availability

The data presented in this study are available in this article.

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
