# Peer review of "Oral Administration of Lactobacillus rhamnosus Ameliorates the Progression of Osteoarthritis by Inhibiting Joint Pain and Inflammation"

_cells, 2021, doi:10.3390/cells10051057_

Round 1
Reviewer 1 Report
I read with interest the article "Oral administration....." by Jhun et al.
Please consider making following changes :
1. Line 332 -
Several strains of probiotics have been studied to date IN OSTEOARTHRITIS.
2. Line 332-335 : Consider briefly mentioning the results of studies 9,10 and 11 that you have quoted as to whether they were positive etc. A single line for each study will be sufficient.
3. Authors should consider mentioning potential limitation or bias in their study.
Overall its an engaging article with robust methodology that purposefully questions our understanding of the subject.
Presentation of result is clear and readability is good.
Author Response
Reviewer 1
I read with interest the article “Oral administration…..” by Jhun et al.
Please consider making following changes:
Line 332-
Several strains of probiotics have been studied to date IN OSTEOARTHRITIS.
Answer : Thanks for comments. We added the above mentioned sentence in Line 346. (marked by red color, page__11_, line__346__).
Line 332-335 : Consider briefly mentioning the results of studies 9,10 and 11 that you have quoted as to whether they were positive etc. A single line for each study will be sufficient.
Answer : Thanks for valuable comment. We added results of studies 9,10, and 11 in discussion (Line 348-353). (marked by red color, page_ 11_, line_348-353_).
Authors should consider mentioning potential limitation or bias in their study. Overall its an engaging article with robust methodology that purposefully questions our understanding of the subject. Presentation of result is clear and readability is good
Answer : Thanks for helpful suggestion. We agree with the referee’s concern. We added our results potential limitation or bias in discussion session. (marked by red color, page_11_, line__373-374_) (marked by red color, page_11_, line__389 )

Reviewer 2 Report
Intestinal destruction in the last sentence of the abstract is a harsh statement...tone this down!
In order to dissuade any bias was the histological and immunohisto-chemistry analysis on the rat joint performed by an independent researcher? Clarify.
Also when the authors say intestine...was it the small bowel or the large bowel that was investigated histologically/immunohistologically?
In Figure 3C it would be useful to further clarify the y - axes designations on the bar graphs, for example the structure bar graph...is the increase in structure does this mean less joint OA structure or is it better joint structure? Clarify.
The discussion should begin with clear statements as to what the present study has demonstrated...there is too much emphasis on further introductory remarks.
Author Response
Reviewer 2
Intestinal destruction in the last sentence of the abstract is a harsh statement…tone this down!
Answer : We agree with the referee’s concern. We revised the word destruction to damage. (marked by red color, page__1_).
In order to dissuade any bias was the histological and immnunohisto-chemistry analysis on the rat joint performed by an independent researcher? Clarify.
Answer : Thank you for your comment. Histological and Immunohistochemistry evaluations were performed independently by two experienced researchers who were blinded to the study groups. Arthritis was induced in Wistar rat (n=6). The experiment was performed three times. The joint tissue was statistically analyzed the joint tissues of three representative animals. The figure 3,4 images were the representative picture.
Also when the authors say intestine… was it the small bowel or the large bowel that was investigated histologically/immunohistologically?
Answer : Thanks for your comment. It was small intestine. We added the abstract, method and result session except for mentioned. (marked by red color, page___1___). (marked by red color, page___3___, line__137__), (marked by red color, page___4___, line__156__), (marked by red color, page___8___, line__294__), (marked by red color, page___9___, line__303, 306__).
In Figure 3C it would be useful to further clarify the y-axes designations on the bar graphs, for example the structure bar graph…is the increase in structure does this mean less joint OA structure or is it better joint structure? Clarify.
Answer : We totally agree with referee’s comment. We clearly corrected the Y-axes designations. The changes figure 3C is as follows.
The discussion should begin with clear statements as to what the present study has demonstrated…there is too much emphasis on further introductory remarks.
Answer : Thank you for these comment. The authors agree with your comments. We revised the part and added suggestion in discussion. (marked by red color, page___10___, line__331-334__).

Reviewer 3 Report
In this manuscript the authors argue that L. rhamnosus can improve joint pain and inflammation, which the authors demonstrate through several rat models of MIA induced osteoarthritis. The authors in the introduction and discussion hypotheses that L. rhamnosus can positively alter the composition of the gut microbiome. However, this hypothesis and cited articles related to this all refer to probiotics, i.e. live bacteria. In the methods, page 13, lines 110-111: “L. rhamnosus (LR-2) was suspended in phosphate-buffered saline (PBS) at 500 mg/kg and killed by heating at 80°C for 30 minutes”. Thus the described experiments used dead L. rhamnosus not living, ergo it is not probiotic. Thus this raises several important questions:
- All references to the mechanisms of action of rhamnosus refer to living bacteria, how do the authors explain their results with dead L. rhamnosus ?
- Thus if the rats were given dead rhamnosus, could the authors explain how this dead bacteria alter the gut microbiome composition in the MIA rats?
- How does dead rhamnosus modulate levels of inflammatory mediators? Or regulate intestinal inflammation in the MIA rats?
- In the human OA chondrocyte experiments alive rhamnosus was used? Thus how do the authors explain the results between human and rat tissue if one used living bacterium and the other dead?
Next, to these questions sever other important discrepancies need to be addressed, particularly several contradictions between the methods and the results:
- In the methods is says (page 13, lines 112-113) that the rats were sacrificed on day 21 after MIA injection, and that rats were injected with MIA injection on day o (Methods, page 13, lines 119-120), However, in the results figure 1 times the rats from day 0 until day 28, a week after they would have been sacrificed. Could the authors explain this serious discrepancy?
- From the methods it is very unclear how many rats were used (in total), which groups of rats exist and how many rats were in each group. Piecing this together from the methods and results it seems that the following 4 groups exist: animal were randomly assigned to the following: Sham MIA injection(saline) fed with (dead) rhamnosus(1) or with a saline solution(2) , MIA induced OA rats fed with (dead) l. rhamnosus(3) or with a saline solution(4). However, in figure 1 only the results of 2 groups, which from the description I think are group 2 (Shamm MIA, saline Fed) and 4 (MIA rats fed (dead) l. rhamnosus ) are shown, not the results from the other groups. Why are only these results shown? Without the other groups the results can not be interpreted.
- Also Figure 1: the methods describe that the weight bearing experiment is performed for both paws (also the paw not effected by OA). Where are the results from the non OA paw? This is an important control?
- Similar, in figure 5 the results of the sham MIA induced rats are missing. Please explain why the controls are missing.
- For the primary human chondrocytes it is not described where the samples were taken from (which joint site), and is the samples were taken from OA lesioned cartilage of OA preserved cartilage, or from joint replacement surgery or from biopsies. Please clarify this in the methods.
Other comments
- Species of the used rats is only mentioned in the abstract, not in the methods, please include such vital information in the methods.
- Can the authors explain why in figure 5 the vehicle treated MIA OA induced rats have such destroyed villi? They rats had induced OA in their joint, not induced inflammation in the gut, thus why would their gut look so drastically different from WT?
- In the introduction sentence: (76-77)” Lactobacillus rhamnosus is known to produce butyrate, a short chain fatty acid (SCFA), through alteration of the gut microbiota.” This is a strange sentence. Please explain how rhamnosus can alter the gut microbiome in order to produce butyrate and provide references for this important claim.
- Discussion: please explain how identifying more therapeutic bacterial strains will elucidate the mechanism of action, if the mechanism of action of current possible probiotics is not known?
Author Response
Reviewer 3
In this manuscript the author that L. rhamnosus can improve joint pain and inflammation, which the authors demonstrate through several rat models of MIA induced osteoarthritis. The authors in the introduction and discussion hypotheses that L. rhamnosus can positively later the composition of the gut microbiome. However, this hypothesis and cited articles related to this all refer to probiotics, i.e. live bacteria. In the methods, page 13, lines 110-11: “ L. rhamnosus was suspended in phosphate-buffered saline (PBS) at 500mg/kg and killed by heating at 80℃ for 30mintes”, Thus this raises several important questions:
- All references to the mechanisms of action of rhamnosus refer to living bacteria, how do the authors explain their results with dead L. rhamnosus?
Answer : Thank you for your comments. Anaerobic bacteria, Lactobacillus rhamnosus, were predicted that the viability of the bacteria would not be constant in the process of delivery to mice after being prepared in a live bacteria form. In this situation, similar functions and mechanisms to live bacteria were reported for heat-killed bacteria as follows, so we selected and injected heat-killed Lactobacillus rhamnosus into MIA rats.
Several reports described about heat-killed bacteria. A number of studies have used inactivated (heat-killed) bacteria to confer a benefit on humans or animals. Using of inactivated bacteria has several benefits; (i.) longer shelf life, (ii.) no risk of infection in vulnerable individuals, (iii.) no loss of activity when used in conjunction with antibiotics or anti-fungal agents (Taverniti and Guglielmetti, Genes Nutr, 2011; de Almada et al., Trands Food Sci, 2016). Several studies revealed that heat-killed bacteria have similar effect compared with live bacteria (Peng et al., Pediatr Alletgy Immunol, 2005; Li et al, Pediatric Res, 2009; Hsieh et al., Food Funct, 2016) and heat-killed bacteria promotes butyrate production in in vitro an in vivo (Kumar et al., Am J Physiol Gastrointest Liver Physiolo, 2015; Canani et al., Appl Environ Microbiol, 2017).
- Thus if the rate sere given dead rhamnosus, could the authors explain how this dead bacteria alter the gut microbiome composition in the MIA rats?
Answer : Thank you for a very interesting comment. We are currently studying live L. acidophilus and dead L. acidophilus in an osteoarthritis MIA model. The diversity was increased in the dead L. acidophilus oral administration group. Also, the propionate producing bacteria was increased the dead L. acidophilus oral administration group. We observed several species, namely Akkermansia, Blautia, Roseburia and Ruminococcaceae, all genus of which were reported to associate with propionate production in intestine. As mentioned above, this study needs the microbiome analysis in dead L. rhamnosus oral administration.
- How does dead rhamnosus modulate levels of inflammatory mediators? Or regulate intestinal inflammation in the MIA rats.
Answer : Thank you for your comments. Dead L.rhamnosus modulation levels of inflammatory mediators. It is thought to be achieved through metabolites like SCFA produced by L. rhmanosus. Short chain fatty acids modulate inflammation by immune cell cytokine production. For example, butyrate and propionate decrease LPS-induced TNFα and nitric oxide synthase (NOS) expression in monocytes. (Vinolo et al., 2011b). These effects are mediated by activation of FFA2 and FFA3 receptors and GPR109A or inhibition of HDACs. As mentioned below, there are many other studies.
1.Short-chain fatty acid butyrate induces IL-10-producing B cells by regulating circadian-clock-related genes to ameliorate Sjögren's syndrome Journal of Autoimmunity 119(2021)102611
2. Alteration Attenuation of Rheumatoid Inflammation by Sodium Butyrate Through Reciprocal Targeting of HDAC2 in Osteoclasts and HDAC8 in T Cells Frontiers in Immunology 2019.9 1525
3. From Dietary Fiber to Host Physiology:Short-Chain Fatty Acids as Key Bacterial Metabolites Cell165, June 2, 2016
- In the human OA chondrocyte experiments alive rhamnosus was used? Thus how do the authors explain the results between human and rat tissue if one used living bacterium and the other dead?
Answer : Thank you for your comment. We used the dead L. rhamnosus. Because of anaerobic bacteria, L. rhamnosus were predicted that the viability of the bacteria would not be constant in the chondrocyte experiments. We used dead rhamnosus in vivo and in vitro experiments.
Next, to these questions sever other important discrepancies need to be addressed, particularly several contradictions between the methods and the results:
- In the methods is says (page13, lines 112-113) that the rats were sacrificed on date 21 after MIA injection, and that rats were injected with MIA injection on day o (Methods, page 13, lines 119-120), However, in the results figure 1 times the rats from day 0 until day 28, a week after they would have been sacrificed. Could the authors explain this serious discrepancy?
Answer : We have miswrote the sentence, and thanks for informing the mistake. We revised the sentence. The rats were sacrificed on day 28 after MIA injection. (marked by red color, page___3___, line__114__).
- From the method it is very unclear how many rats were used (in total), which groups of rats exist and how many rats were in each group. Piecing this together from the methods and results it seems that the following 4 groups exist: animal were randomly assigned to the following: Sham MIA injection(saline) fed with(dead) rhamanosus (1) or with a saline solution (2), MIA induced OA rats fed with (dead) rhamanosus(3) or with a saline solution(4). However, in figure 1 only the results of 2 groups, which from the description I think are group 2 (Sharmm MIA ,saline Fed) and 4 (MIA rats fed (dead)/, rhamnosus) are shown, not the results from the other groups. Why are only these results shown? Without the other groups the results can not be interpreted.
Answer : I am very sorry to confuse referee. Thanks for critical comment. We have two groups. There are MIA induced OA rats fed with (dead) rhamnosus (1) and MIA induced OA rats fed with (vehicle) saline solution. Also, Arthritis was induced in Wistar rat (n=6). The experiment was performed three times.
- Also Figure 1: The methods describe that the weight bearing experiment is performed for both paws (also the paw not effected by OA). Where are the results from the non OA paw? This is an important control?
Answer : We have miswrote the sentence, and thanks for informing the mistake. We delete miswrote sentence in method section. We have MIA induced OA rats fed with (dead) rhamanosus (1) and MIA induced OA rats fed with (vehicle) saline solution. (marked by red color, page___3___, line__109-113__).
- Similar, in figure 5 the results of the sham MIA induced rats are missing. Please explain why the controls are missing.
Answer : Thanks for critical comment. We have only MIA induced OA rats fed with (dead) rhamanosus (1) and MIA induced OA rats fed with (vehicle) saline solution.
- For the primary human chondrocytes it is not described where the sample were taken from (which joint site), and is the samples were taken from OA lesioned cartilage of OA preserved cartilage, or from joint replacement surgery of from biopsies. Please clarify this in the methods.
Answer : We totally agree with referee’s comment. We revised the method section. (marked by red color, Page _4_, line 180-182_).
Other comments
- Species of the used rats is only mentioned in the abstract, not in the methods, please include such vital information in the methods.
Answer : Thanks for your comment. We revised the method section. (marked by red color, Page _3_, line 118_).
- Can the authors explain why in figure 5 the vehicle treated MIA OA induced rats have such destroyed villi? They rats had induced OA in their joint, not induced inflammation in the gut, thus why would their gut look so drastically different from WT?
Answer : We agree with the referee’s concern. Currently, there are many studies on the gut joint axis. There are reports that when an dysbiosis in the intestine occurs, not only arthritis but also various disease occurs. But, there is no research yet on cases of intestinal problems and dysbiosis when arthritis occurs. We would like to proceed with research on the reason. Interestingly, in our study, the rheumatoid arthritis model also collagen induced arthritis, the intestine was damaged and the length of the intestine was shortened. In the future, we will take this part into account and try to confirm it in subsequent studies. Thank you very much for your review comment.
- In the introduction sentence: (76-77)” Lactobacillus rhamnosus is known to produce butyrate, a short chain fatty acid (SCFA), through alteration of the gut microbiota.” This is a strange sentence. Please explain how rhamnosus can alter the gut microbiome in order to produce butyrate and provide references for this important claim.
Answer : We agree with the referee’s concern. In the previous study Lactobacillus rhamnosus GG-supplemented formula expands butyrate-producing bacterial strains. (ISME J 2016, 10, 742-750). As well as, Butyrate is a natural fermentation product of the gut, and it plays a crucial role in maintaining the homeostasis for host metabolism and gut microbiome diversity in inflammatory bowel disease (IBD) anima model. (Animals 2020, 10(7), 1154). In addition to this paper, there are reports that butyrate increases the diversity of the microbiome. But so far, these study it is not possible to determine the mechanism by which L. rhamnosus treatment influence microbial community.
- Discussion: please explain how identifying more therapeutic bacterial strains will elucidate the mechanism of action, if the mechanism of action of current possible probiotics is not known?
Answer : Thank you for a very interesting comment. In general, in order to investigate the mechanism of probiotics for the treatment of diseases, it is a common method to investigate changes in the microbiome and increase and decrease of short chain fatty acid according to taking probiotics, and what mechanism is involved in the therapeutic effect. Although the therapeutic effect of L. rhamnosus in many diseases has been identified, it is necessary to study changes such as microbiome change and SCFA for administration of L. rhamnosus in disease. In the future study, we will take this part into account and try to confirm it in subsequent studies. Thank you very much for your review comment.

Round 2
Reviewer 3 Report
Comments to the authors(2)
I thank the authors for their clear answers and their improvements made to their article. I believe this to be an interesting study, however I still find that the way the study is described to be unclear at best or disingenuous at worst. I think the authors should and need change these things in order to accurately and fairly present their research to the reader.
Currently the study still presents itself as a study toward a probiotic treatment with L.rhamnosus, which is incorrect as they use heat-killed bacteria. Although there indeed is evidence to suggest effectiveness of heat-killed bacteria, this is not consensus knowledge nor included under the definition of probiotics. In the introduction all studies cites make use of live bacteria, no mention is given on heat-killed bacteria nor are any of such studies referred to (there are many which the authors could have cited[1]). This gives the wrong impression of the study and of the biology behind the mechanism of action (there is a subtle, but definite difference in mechanism of action with living or dead bacteria)
[1] Piqué N, Berlanga M, Miñana-Galbis D. Health Benefits of Heat-Killed (Tyndallized) Probiotics: An Overview. Int J Mol Sci. 2019;20(10):2534, 2019
In continuation, within the discussion the authors, unfortunately, also do not address the heat-killed bacteria, the current understanding of their mechanism of action of heat-killed bacteria[1] or the benefits that such a treatment can have over common probiotic treatment [1]. Also the authors in their manuscript do not address the (interesting/important) result of the very badly damaged villi of the vehicle treated MIA OA induced rats (figure 5). This result/observation need to be addressed in the results/discussion.
This is in my professional option that key elements of their study design absolutely need to be central and addressed in their manuscript. To not do so, is disingenuous. Please change this in the manuscript.
In addition, the methods remain unclear despite the authors giving clear answers to these points in their response. Please make the methods explicitly clear.
- It is not clear from the text that there are two groups of rats or how many rats in total were used.
- It is not clear from the text that also heat-killed rhamnosus was also used in the human cell culture experiments
- The authors now describe that the cartilage came from individuals with OA undergoing total joint replacement, but from which joint was the cartilage taken? Hip or knee or both?
In addition, I still have several questions regarding some results:
- In the methods the weight bearing experiment was performed on both paws in the rats (both paws: left and right?), which paw is shown in figure 1 and why are not both paws shown?
- Can the authors explain why there are no SHAM MIA induced rats included in this study as a control?
Author Response
Answers: We really appreciate your time and effort to edit our manuscript. In this revised manuscript, we have resolved most of the issues raised by the reviewers as you can see in our response to their comments below. We think the reviewer’s suggestions have tremendously improved the quality of this manuscript and we hope the revised manuscript fulfils the requirements for its publication in Cells. Detailed responses the reviewer’s critique is elaborated below.
I thank the authors for their clear answers and their improvements made to their article. I believe this to be an interesting study, however I still find that the way the study is described to be unclear at best or disingenuous at worst. I think the authors should and need change these things in order to accurately and fairly present their research to reader.
Currently the study still presents itself as a study toward a probiotic treatment with L.rhamnosus, which is incorrect as they use heat-killed bacteria. Although there indeed is evidence to suggest effectiveness of heat-killed bacteria, this is not consensus knowledge nor included under the definition of probiotics.
In the introduction all studies cites make use of live bacteria, no mention is given on heat-killed bacteria nor are any of such studies referred to (there are many which the authors could have cited[1]). This gives the wrong impression of the study and of the biology behind the mechanism of action (there is a subtle, but definite difference in mechanism of action with living or dead bacteria)
[1] Piqué N, Berlanga M, Miñana-Galbis D Health Benefits of Heat-Killed (Tyndallized) Probiotics: An Overview. Int J Mol Sci.2019;20(10):2534, 2019 In continuation, within the discussion the authors, unfortunately, also do not address the heat-killed bacteria, the current understanding of their mechanism of action of heat-killed bacteria[1] or the benefits that such a treatment can have over common probiotic treatment [1].
Answers: We thank the reviewer for your constructive and helpful comments concerning the manuscript. We revised the introduction and discussion section (marked by red color, Page 2~3, line 89-102, Page 12, line 384-387).
Also the authors in their manuscript do not address the (interesting/important) result of the very badly damaged villi of the vehicle treated MIA OA induced rats (figure 5). This result/observation need to be addressed in the results/discussion.
Answers: Thanks for valuable suggestion. We revised the discussion section (marked by red color, Page 13, line 406-412).
This is in my professional option that key elements of their study design absolutely need to be central and addressed in their manuscript. To not do so, is disingenuous. Please change this in the manuscript.
Answers: Thanks for valuable and Helpful suggestion. As mentioned above, added the manuscript.
In addition, the methods remain unclear despite the authors giving clear answers to these points in their response. Please make the methods explicitly clear. It is not clear from the text that there are two groups of rats or how many rats in total were used.
Answers: Thank you for your comment. We added the method session. (marked by red color, page_3_, line 116-118).
It is not clear from the text that also heat-killed rhamnosus was also used in the human cell culture experiments.
Answers: We added the method session. (marked by red color, page__5__, line_219_).
The authors now describe that the cartilage came from individuals with OA undergoing total joint replacement, but from which joint was the cartilage taken? Hip or knee or both?
Answers: Thank you for your comment. We added the method session. (marked by red color, page__5__, line__199-200__).
In addition, I still have several questions regarding some results.
In the methods the weight bearing experiment was performed on both paws in the rats (both paws: left and right?), which paw is shown in figure 1 and why are not both paws shown?
Answers: I am very sorry to confuse referee. The weight bearing was missing in the method. The weight bearing was missing in the method. We added the method session. (marked by red color, page___3-4___, line__145-152__). The principle of weight bearing is to conduct experiments with the right and left feet injected with MIA. It uses the principle of putting less force on the sore foot, so if the pain in the foot decreases, it creates a balance of power. This is because the closer to 50, the same force is applied to both feet
Can the authors explain why there are no SHAM MIA induced rats included in this study as a control?
Answers: We really appreciate your time and effort to edit our manuscript. In the last experiment, we performed the sham MIA experiment. Control rats were injected with an equivalent volume of saline. The H&E and safranin O stain was similar to that of normal tissue as shown below. Therefore, the SHAM MIA experiment was not progressed in this study.
